# Host genetic factors related to innate immunity, environmental sensing and cellular functions are associated with human skin microbiota

Despite the increasing knowledge about factors shaping the human microbiome, the host genetic factors that modulate the skin-microbiome interactions are still largely understudied. This contrasts with recent efforts to characterize host genes that influence the gut microbiota. Here, we investigated the effect of genetics on skin microbiota across three different skin microenvironments through meta-analyses of genome-wide association studies (GWAS) of two population-based German cohorts. We identified 23 genome-wide significant loci harboring 30 candidate genes involved in innate immune signaling, environmental sensing, cell differentiation, proliferation and fibroblast activity. However, no locus passed the strict threshold for study-wide significance ($P < 6.3 \times 10^{-10}$ for 80 features included in the analysis). Mendelian randomization (MR) analysis indicated the influence of staphylococci on eczema/dermatitis and suggested modulating effects of the microbiota on other skin diseases. Finally, transcriptional profiles of keratinocytes significantly changed after in vitro co-culturing with *Staphylococcus epidermidis*, chosen as a representative of skin commensals. Seven candidate genes from the GWAS were found overlapping with differential expression in the co-culturing experiments, warranting further research of the skin commensal and host genetic makeup interaction.

Human-associated microbial communities show individual-specific variation shaped by a multitude of factors[1,2]. For skin in particular, the bacterial community composition is strongly influenced by host characteristics, such as skin microenvironment, sex, age and body mass index (BMI), and to a lesser extent by lifestyle and environmental expositions[3]. The genetic influence of the host on skin microbiome composition and diversity was suggested by findings indicating heritability of up to 56.4% for single taxonomic branches of skin commensals in twins[4]. Furthermore, host genetics and skin microbiota interactions haven been suggested by studies including targeted genes[4] and in the context of inflammatory diseases, such as atopic dermatitis[5].

Nevertheless, the influence of host genetics on the skin microbiome is largely understudied and no dedicated genome-wide association study (GWAS) of host genetics and the bacterial community inhabiting the skin has been performed so far. This strongly contrasts with what is known about the human gut microbiota, where a variety of associated genomic loci and pathways has been identified by large GWAS[6,7]. Together, these gut microbiome-based GWAS have not only suggested how human molecular mechanisms modulate the microbiome but also indicated the consequences of such modulation to the host health and disease.

Therefore, we aimed to study the effects of genetics on skin microbiota across skin microenvironments through meta-analyses of

✉ e-mail: a.franke@mucosa.de; sweidinger@dermatology.uni-kiel.de

GWAS of two German cohorts. To investigate the putative influence of the skin microbiota in skin diseases we applied Mendelian randomization (MR) analysis. Finally, putative effects of the skin microbiome members on the expression of candidate genes identified by GWAS were tested using normal human epidermal keratinocytes cultured with the common skin bacterium *Staphylococcus epidermidis*.

## Results and discussion

A total of 1656 skin samples from participants of two cross-sectional, population-based German cohorts, KORA FF4 ($n_{Individuals}$ = 324) and PopGen ($n_{Individuals}$ = 273)[8,9] were analyzed. Skin samples were taken from dry [dorsal and volar forearm (PopGen)], moist [antecubital fossa (KORA FF4 and PopGen)] and sebaceous [retroauricular fold (KORA FF4) and forehead (PopGen)] skin microenvironments (Fig. 1a–c, Supplementary Table 1). Microbial community profiles were obtained from sequencing of the V1-V2 regions from the 16 S ribosomal RNA (rRNA) gene (see Methods). Genome-wide association analyses were conducted on univariate relative abundances of individual bacteria (amplicon sequence variants; ASVs) and non-redundant taxonomic groups ranging from genus to phylum levels (79 in total; see Methods). Additionally, multivariate community composition (i.e., beta diversity as captured by Bray-Curtis dissimilarity) was analyzed for association with host genetic variation. The umbrella term "microbial feature" will henceforth be used in this article for all 80 analyzed input data.

We tested the association of microbial features with variation in 4,685,714 human autosomal single nucleotide polymorphisms (SNP), accounting for main confounders of the skin microbiota (age, sex and BMI) and genetic background of study participants (see Methods)[3,7]. Cohort-wise association results were combined in a meta-analysis framework according to skin microenvironment, justified by the observed similarity of the microbiota profiles of samples from the same microenvironment (Fig. 1d). To assure robustness of association results, only loci with genome-wide significance ($P_{Meta}$ < 5 × 10$^{-8}$) and with nominal significance in both cohorts ($P$ < 0.05) were further considered (see Methods for details).

A total of 23 loci showed a genome-wide significant association with skin microbial features, of which 22 were linked to univariate features (Table 1 and Fig. 2a). However, none of these passed the strict threshold for study-wide significance ($P$ < 6.3 × 10$^{-10}$ for 80 features

included in the analysis, see Methods). Most of the associations were found in moist skin microenvironment ($n$ = 11), followed by dry ($n$ = 7) and sebaceous ($n$ = 5) (Fig. 2b). There was a tendency for a higher number of associations found in deeper taxonomic levels: the highest number of significant associations were found at the ASV level ($n$ = 8), followed by genus level ($n$ = 6; Fig. 2c). Of all microbial features deeper than family level, features within the genus *Staphylococcus* were associated with most loci ($n$ = 5; Fig. 2d). Bayesian fine-mapping or linkage disequilibrium (LD) structure prioritized 462 genetic variants as potentially causal (Supplementary Data 1). A total of 30 genes were found of interest for containing potentially causal variants and/or because these variants were significantly associated with the gene expression in skin tissue from the GTEx portal[10] (Table 1). Of these, 27 were protein coding genes, one an rRNA pseudogene and two long-non coding RNA (lncRNA) genes (Supplementary Data 2). Most of the protein coding genes were expressed in skin tissue and found expressed in different cell types in skin in datasets from previous studies (see Methods for details[11,12]) (Fig. 3). In the next section, we will explore the genes of interest with functional roles related to the host-microbiome interface.

### Host functions associated with the human skin microbiota

Genetic variants associated with the skin bacteria were localized in genes related to pathogen sensing and regulation of response to pathogens. *C1QBP* (locus id: 22, lead variant rs2472614, $P_{Meta}$ = 4.7 × 10$^{-8}$, associated with ASV086 [*Acinetobacter johnsonii*]), for instance, encodes the complement component 1, q subcomponent binding protein (C1qBP, a.k.a. gC1q-R/p33) and is abundantly expressed in keratinocytes (Fig. 3[12]). C1qBP is an ubiquitous, multi-ligand, multi-functional and multicompartmental protein, which also acts as endo-thelial receptor to plasma proteins from the complement and kinin/kallikrein systems and is a marker for epithelial cell proliferation[13,14]. C1qBP binds to microbial proteins[15], including *Staphylococcus aureus* protein A[16], and therefore, is suggested to play a role in both the response to and pathogenesis of microbes[17]. Additionally, *DHX33*, (locus id: 22, same locus containing gene *C1QBP*), and *CARD8* (locus id: 23, rs6509364, $P_{Meta}$ = 8.5 × 10$^{-09}$, associated with the *Rhodobacteraceae* family) encode proteins which regulate inflammasome activity, which in turn regulate innate immunity caspase 1 activation[18]. DHX33

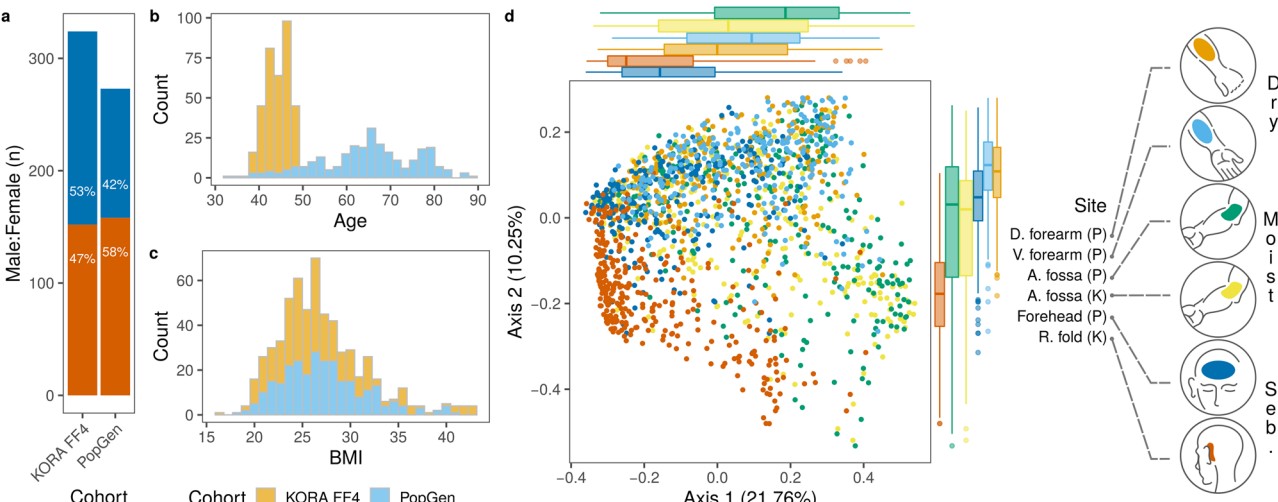

**Fig. 1 | Characteristics of KORA FF4 and PopGen cohorts. a** Female (orange) and male (blue) composition of cohorts. **b** age and, **c** body mass index (BMI) distribution in cohorts. **d** ordination of skin microbiome profiles based on Bray-Curtis dissimilarity and principal coordinates analysis. Samples ($n$ = 1,656) were colored by the skin site and represent dry [dorsal (D.) forearm ($n$ = 260) and volar (V.) forearm ($n$ = 251)], moist [antecubital (A.) fossa ($n$ = 318 in KORA FF4, $n$ = 258 in PopGen)] and sebaceous [seb.; forehead ($n$ = 252) and retroauricular (R.) fold ($n$ = 317)] microenvironments. Cohort names were abbreviated, PopGen (P) and KORA FF4 (K). Marginal boxplots are shown to visualize sample distributions along axes. The boxplot area represents the interquartile range (IQR) divided by the median. Lines extend to a maximum of 1.5 × IQR beyond the area. Points are outliers. Percentage of variation explained by each axis is shown in parentheses.

## Table 1 | Results summary of GWAS for microbial features

| ID | Chr | Position | rsID | Effect allele | EAF | Other allele | Microenv. | Feature | N (total) | Beta ± s.e. | P value | Genes |
|---|---|---|---|---|---|---|---|---|---|---|---|---|
| 1 | 2 | 17323400 | rs1396075 | T | 0.76 | A | Moist | c.Gammaproteobacteria | 563 | 0.37 ± 0.067 | $3.4 \times 10^{-08}$ | ■ |
| 2 | 2 | 43812831 | rs12466030 | A | 0.66 | G | Moist | a.ASV070 [*Veillonella* (unc.)] | 226 | 0.583 ± 0.103 | $1.7 \times 10^{-08}$ | THADA |
| 3 | 3 | 9164097 | rs2664121 | T | 0.73 | G | Dry | g.*Micrococcus* | 402 | −0.33 ± 0.112 | $4.3 \times 10^{-09}$ | *ENSG00000269886*,SRGAP3 |
| 4 | 3 | 12514124 | rs709165 | G | 0.54 | A | Moist | a.ASV006 [*S. hominis*] | 398 | 0.379 ± 0.069 | $4.0 \times 10^{-08}$ | **MKRN2**,MKRN2OS,RAF1,RNA5SP123,**TSEN2** |
| 5 | 4 | 3266916 | rs2159173 | A | 0.93 | T | Sebaceous | a.ASV093 [*Staphylococcus* (unc.)] | 276 | −0.957 ± 0.159 | $1.8 \times 10^{-09}$ | HTT,MSANTD1,RGS12 |
| 6 | 4 | 55057749 | rs55702239 | G | 0.77 | A | Dry | o.Bacteroidales,g.Bacteroides | 349 | −0.43 ± 0.103 | $3.5 \times 10^{-08}$ | FIP1L1,PDGFRA |
| 7 | 5 | 14584609 | rs152620 | A | 0.79 | T | Moist | g.*Acinetobacter* | 454 | −0.444 ± 0.081 | $3.7 \times 10^{-08}$ | OTULINL |
| 8 | 6 | 69060156 | rs9445997 | T | 0.62 | C | Sebaceous | g.*Staphylococcus* | 569 | −0.328 ± 0.06 | $4.0 \times 10^{-08}$ | ■ |
| 9 | 6 | 93109029 | rs2757026 | C | 0.51 | T | Moist | f.Clostridiales_Incertae_Sedis_XI | 363 | −0.404 ± 0.072 | $2.2 \times 10^{-08}$ | ■ |
| 10 | 6 | 144022040 | rs9484795 | T | 0.77 | C | Dry | g.*Anaerococcus* | 352 | −0.41 ± 0.132 | $3.4 \times 10^{-08}$ | **PHACTR2** |
| 11 | 7 | 57369680 | rs11762959 | A | 0.55 | G | Moist | o.Lactobacillales | 489 | 0.348 ± 0.064 | $4.5 \times 10^{-08}$ | ■ |
| 12 | 7 | 57369974 | rs7791487 | A | 0.56 | T | Moist | f.Streptococcaceae | 458 | 0.378 ± 0.064 | $3.4 \times 10^{-09}$ | ■ |
| 13 | 8 | 125763112 | rs59379063 | T | 0.92 | A | Sebaceous | a.ASV093 [*Staphylococcus* (unc.)] | 279 | −0.733 ± 0.133 | $3.2 \times 10^{-08}$ | ■ |
| 14 | 9 | 12691926 | rs10121400 | T | 0.64 | A | Moist | o.Burkholderiales | 509 | −0.354 ± 0.063 | $2.4 \times 10^{-08}$ | ■ |
| 15 | 11 | 106233170 | rs17105612 | G | 0.94 | A | Moist | a.ASV013 [*S. epidermidis*] | 361 | −0.928 ± 0.165 | $1.8 \times 10^{-08}$ | ■ |
| 16 | 12 | 96938142 | rs12423627 | T | 0.93 | C | Sebaceous | a.ASV002 [*Staphylococcus* (unc.)] | 568 | 0.653 ± 0.113 | $6.9 \times 10^{-09}$ | CFAP54 |
| 17 | 12 | 101194600 | rs4764996 | G | 0.93 | A | Dry | a.ASV013 [*S. epidermidis*] | 348 | −1.2 ± 0.204 | $2.1 \times 10^{-08}$ | ANO4 |
| 18 | 13 | 33581388 | rs1543797 | T | 0.75 | C | Dry | Beta-diversity | 511 | ■ | $3.2 \times 10^{-08}$ | ■ |
| 19 | 13 | 38067575 | rs12583353 | A | 0.89 | G | Dry | g.*Paracoccus* | 372 | −0.94 ± 0.144 | $5.6 \times 10^{-10}$ | ■ |
| 20 | 14 | 33629140 | rs17100281 | G | 0.94 | A | Moist | a.ASV021 [*Micrococcus* (unc.)] | 331 | −0.969 ± 0.176 | $3.7 \times 10^{-08}$ | NPAS3 |
| 21 | 16 | 28056516 | rs80490083 | C | 0.72 | A | Sebaceous | a.ASV004 [*Corynebacterium* (unc.)] | 505 | −0.378 ± 0.068 | $2.8 \times 10^{-08}$ | GSG1L |
| 22 | 17 | 5341050 | rs2472614 | G | 0.89 | C | Dry | a.ASV086 [*A. johnsonii*] | 255 | −1.08 ± 0.186 | $4.7 \times 10^{-08}$ | C1QBP,DERL2,DHX33,**ENSG00000263272**,MIS12,NUP88,RABEP1,**RPAIN** |
| 23 | 19 | 48742067 | rs6509364 | C | 0.65 | T | Moist | f.Rhodobacteraceae | 377 | −0.415 ± 0.072 | $8.5 \times 10^{-09}$ | CARD8,TMEM143,**ZNF114** |

Single variant association tests were performed for each sample type and each microbial feature. Tests were adjusted for age, sex, body mass index (BMI) and genetic background (first ten genetic principal components). Positions are given in genome assembly hg19 (GRCh37). Effect allele frequency (EAF) and total sample number (N) for the meta-analysis (sample pairs for dry) are shown. Results from moist and sebaceous skin sites were combined by meta-analysis (using METAL software for beta diversity and METASOFT software for univariate microbial features) and considered significant when $P_{Meta}$ value were genome-wide significant ($P_{Meta} < 5 \times 10^{-8}$) and data sets were considered significant when at least one data set resulted in genome-wide significance (lowest P value shown as $P_{Meta}$) and the other in nominal significance. Meta-analyses were weighted by sample size for multivariate microbial feature and by inverse variance for univariate features. Effect sizes (β) and its standard error (s.e.) from meta-analyses are shown for moist and sebaceous. Effect size and standard error from tests with volar forearm are shown for dry skin. Tests were two-sided. Candidate causal variants were identified by fine-mapping or based on LD > 0.6 to the lead genetic variant. Genes with variants within their region (no formatted font) or with variants associated with their expression (italic font) are shown. Genes are shown in bold font when both conditions are met. Microbial features are prefixed with their level, amplicon variant sequence (a.), genus (g.), family (f.), order (o.), class (c.) or phylum (p.). Association with rs55702239 in dry sites have been identified with the non-redundant features o.Bacteroidales and g.*Bacteroides*. For simplicity, only statistics related to the genus level are shown in this case. ENSG00000263272 is a novel transcript, antisense to RPAIN and ENSG00000269886 is a novel transcript, antisense to TTLL3.

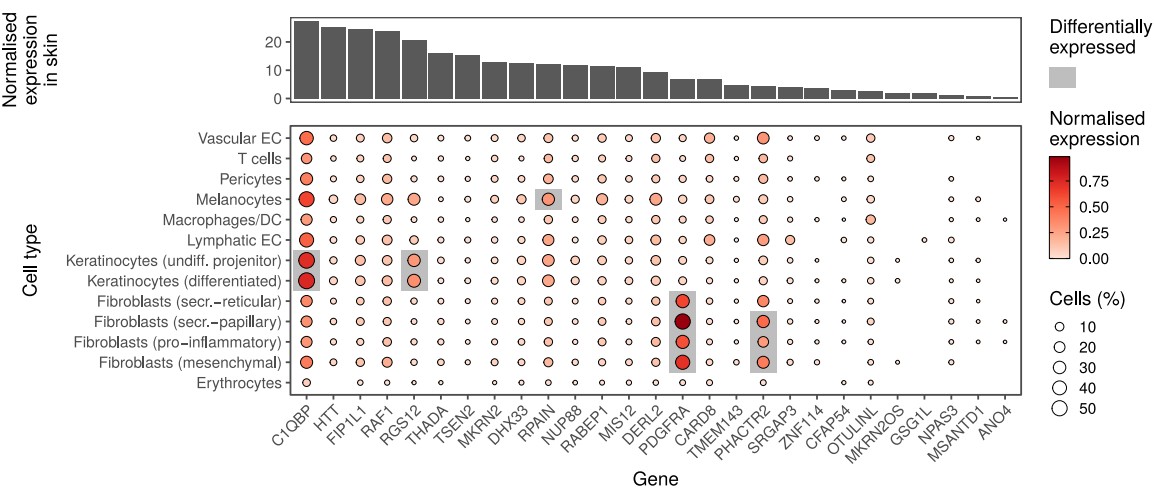

**Fig. 2 | Results from the GWAS. a** Manhattan plot of per skin microenvironment meta-analysis. Lowest *P* value of each position is shown and identified by locus ID and rsID. Meta-analysis *P* values were obtained using the software METAL and METASOFT or by combining *P* values from data sets that originated from dry skin sites, see Methods. Significant positions are colored according to skin microenvironment and listed, where leading genetic variant, protein coding genes selected by fine-mapping as containing possible causal variants and microbial features are reported. Table 1 contains the list of loci characteristics and genes. **b** Count of significantly associated loci per microenvironment. **c** Level of microbial features with highest number of significant associations. **d** Sub-family features with the highest number of significant associations.

**Fig. 3 | Expression of human genes associated with the skin microbiome in public databases.** Candidate protein coding genes were selected by GWAS in skin. Upper panel shows the normalized transcriptional expression of genes in skin tissue. Data are from Human Protein Atlas version 20.1[11], which additionally includes data sets from the GTEx[10] and the Functional annotation of the mammalian genome (FANTOM5)[68] projects. Bottom panel shows candidate gene expression in different skin cell types. Single-cell expression was normalized by cell type. Genes differently expressed in each cell type in comparison with the others are highlighted. Displayed log-normalized gene expression data and differential expression analyses are retrieved from Solé-Boldo et al.[12]. Candidate genes were mapped by gene symbol.

activates the NLRP3 inflammasome after sensing cytosolic RNA derived from viruses, bacteria or achaea[19,20]. CARD8 is structurally related to NLRP1, a sensor component of the NLRP1 inflammasome, and has been shown to activate caspase 1 activity in resting T cells and is a negative regulator of NLRP3 inflamasome[21,22]. Together, these results suggest that innate immune components carrying out sensing and regulatory activities may be involved in shaping the human skin microbiota.

Associated genetic variants were also localized at genes *HTT* (locus id: 5, rs2159173, $P_{Meta} = 1.8 \times 10^{-09}$, associated with ASV093 [*Staphylococcus* (uncl.)]) and *CFAP54* (locus id: 16, rs12423627, $P_{Meta} = 6.9 \times 10^{-09}$, associated with ASV002 [*Staphylococcus* (uncl.)]), which encode proteins required for cilia formation in mammalian cells[23,24] and expressed in different cell types in skin (Fig. 3). Further, we found SNPs that were associated with the expression of the transcript ENSG00000269886 (locus id: 3, rs2664121, $P_{Meta} = 4.3 \times 10^{-09}$, associated with the genus *Micrococcus*). Interestingly, the effector alleles of all of these (rs2664121, rs2075337, rs2543492, rs1300250) were associated with the decrease in both tissue expression of ENSG00000269886 and relative abundance of the genus *Micrococcus* (the GTEx portal[10] and Supplementary Data 1). ENSG00000269886 is an lncRNA antisense to the gene *TTLL3*, which regulates cilia assembly across eukaryotes[25,26]. Skin cells do not have motile cilia. Thus, it is likely that these genes are related to primary cillium, an organelle at the cell surface that senses extracellular signals, such as chemomechanical signals, osmolarity, pH, oxygen and light[27]. Primary cillium is found in various skin cells such as keratinocytes, fibroblasts, melanocytes and Langerhans cells[28]. Its formation is influenced by the dynamics of the actin cytoskeleton[29], which is regulated by *SRGAP3* encoded protein (locus id: 3)[30]. Together, these results suggest that extracellular sensing through primary cilium may be involved in the regulation of the skin microbiota.

Additional associations were observed with SNPs located in genes involved in cellular differentiation and proliferation. These were *RAF1* (locus id: 4, rs709165, $P_{Meta} = 4.0 \times 10^{-08}$, associated with ASV006 [*Staphylococcus hominis*])[31–33] and *RGS12* (locus id: 5, rs2159173)[34–36], the latter found abundantly expressed in keratinocytes[12] (Fig. 3). Furthermore, SNPS in *PDGFRA* (locus id: 6, rs55702239, $P_{Meta} = 3.5 \times 10^{-08}$) were associated with order Bacteroidales and genus *Bacteroides*. *PDGFRA* is abundantly expressed in fibroblasts[12] (Fig. 3) and participates in cellular maintenance[37] and extracellular matrix production[38]. Keratinocyte proliferation, differentiation and function as well as innate immune signaling are major forces contributing to the complex function of the skin barrier. Therefore, it is conceivable that the discovered GWAS associations may represent links between the skin barrier and members of the skin microbiota.

## Expression of candidate genes by keratinocytes co-cultured with *Staphylococcus epidermidis*

To gain insights in the putative participation of the identified candidate genes in the molecular interaction with the skin bacteria, we analyzed the in vitro transcriptional profile of normal human epidermal keratinocytes co-cultured with *S. epidermidis*, an abundant commensal in human skin[39]. Transcriptional profiles (six replicates) of keratinocytes from the foreskin of a 0-year-old male donor co-cultured with the *S. epidermidis* ATCC 14990 strain clearly differed from the profiles of controls, keratinocytes that were not co-cultured with bacteria (Fig. 4a). The *S. epidermidis* ATCC 14990 strain is a well characterized laboratory strain which is close to the strains found in the skin of the participants of the two cohorts studied. This proximity is suggested by the observation of 100% overlap and identity with the full length of ASV002 [*Staphylococcus* (uncl.)] amplicon sequence (307 base pairs), the second most abundant ASV in the whole database (~10% of rarefied sequences) and the most abundant ASV assigned to *Staphylococcus* genus.

A total of 4134 genes were differentially regulated (Supplementary Data 3), suggesting a strong transcriptional response of human keratinocytes to *S. epidermidis* ATCC 14990 strain in vitro. According to pathway enrichment analysis (Supplementary Data 4), the most significant biological processes upregulated were related to immune response, including cytokine-mediated and innate immune responses, as well as response to virus and symbionts (Fig. 4b). On the other hand, ribosomal biogenesis, processing of ribosomal RNA and non-coding RNA were among the most significantly down regulated biological processes. In this scenario, a quarter of the candidate genes ($n = 7$) were differentially expressed ($q < 0.05$ and absolute log2 fold change >1) when comparing cultures with and without *S. epidermidis* ATCC 14990 (Fig. 4c).

Based on knockout mouse macrophage cells, the deficiency of C1QBP protein increases the DNA sensor cyclic GMP-AMP (cGAMP) synthase-induced innate immune response[40]. Here, we observed the downregulation of *C1QBP* transcription associated with the upregulation of genes belonging to innate immune response (Fig. 4b, c), which sides with our GWAS suggestion that this gene may play a role in the regulation of skin bacteria via innate immunity. On the other hand, *DHX33* was downregulated, contrasting to its role in innate immunity via activation of NLRP3, which transcript was upregulated (Fig. 4c and see Supplementary Data 3). It is thus likely that the reduced expression of *DHX33* in our assays may be associated with the role of DHX33 in rRNA synthesis via positive regulation of transcription by RNA polymerase I[41], being both pathways downregulated (Fig. 4b, c, Supplementary Data 4).

Genes coding for SRGAP3 and TTLL3, of which the lncRNA antisense gene was implied by GWAS, were upregulated (Fig. 4c and Supplementary Data 3). These observations support our discovered association of primary cilium and skin bacteria. However, it is important to bear in mind that the encoded proteins are not exclusively related to primary cilium, and their expression in our assays may also be related to other structures, e.g., cytoskeleton in the case of SRGAP3[30], and processes, e.g., proliferation in the case of TTLL3[26]. Finally, the know role of PDGFRA in fibroblast activity are not directly translated to keratinocytes[37,38]. Therefore, the consequences of *S. epidermidis*-induced in vitro upregulation of *PDGFRA* in keratinocytes remain to be investigated.

Our in vitro experiment is explorative in nature and is limited to its reductionist approach: it consists of two-dimensional co-cultures of isolated keratinocytes and a single *S. epidermidis* laboratory strain. It is well known that the immunomodulatory effects of *S. epidermidis* depend on the specific strain, and that there is a large *S. epidermidis* strain level variation. Thus, it is not possible to directly extrapolate our preliminary functional results to an eventual keratinocyte response to skin commensals in vivo. A panel of commensal strains as well as in vitro models closer to the skin physiology, such as three-dimensional human skin models[42], are necessary to uncover the functional dynamics of the host-commensal cellular interactions. Nevertheless, our assays allowed for the observation of the transcriptional regulation of several GWAS selected genes, being a starting point for functional investigations of the roles of these genes in the interaction with the skin microbiota.

## Influence of skin microbiota on non-infectious skin diseases

Summary statistics of univariate microbial features were used as exposures in 2-sample mendelian randomization (MR; see Methods) to assess their influence on non-infectious skin diseases. A total of eight comparisons passed the per-trait suggestive threshold ($q_{(trait)}$ value <0.05, Fig. 5), although no comparison passed the global threshold ($q_{(global)}$ value <0.05; Supplementary Data 5). MR results indicated the influence of staphylococci in dermatitis/eczema (*Staphylococcus* genus, $\beta = 1.5 \times 10^{-03}$), and further, modulating roles of *Flavobacteriaceae* in two allergy-related traits with microenvironment-specific

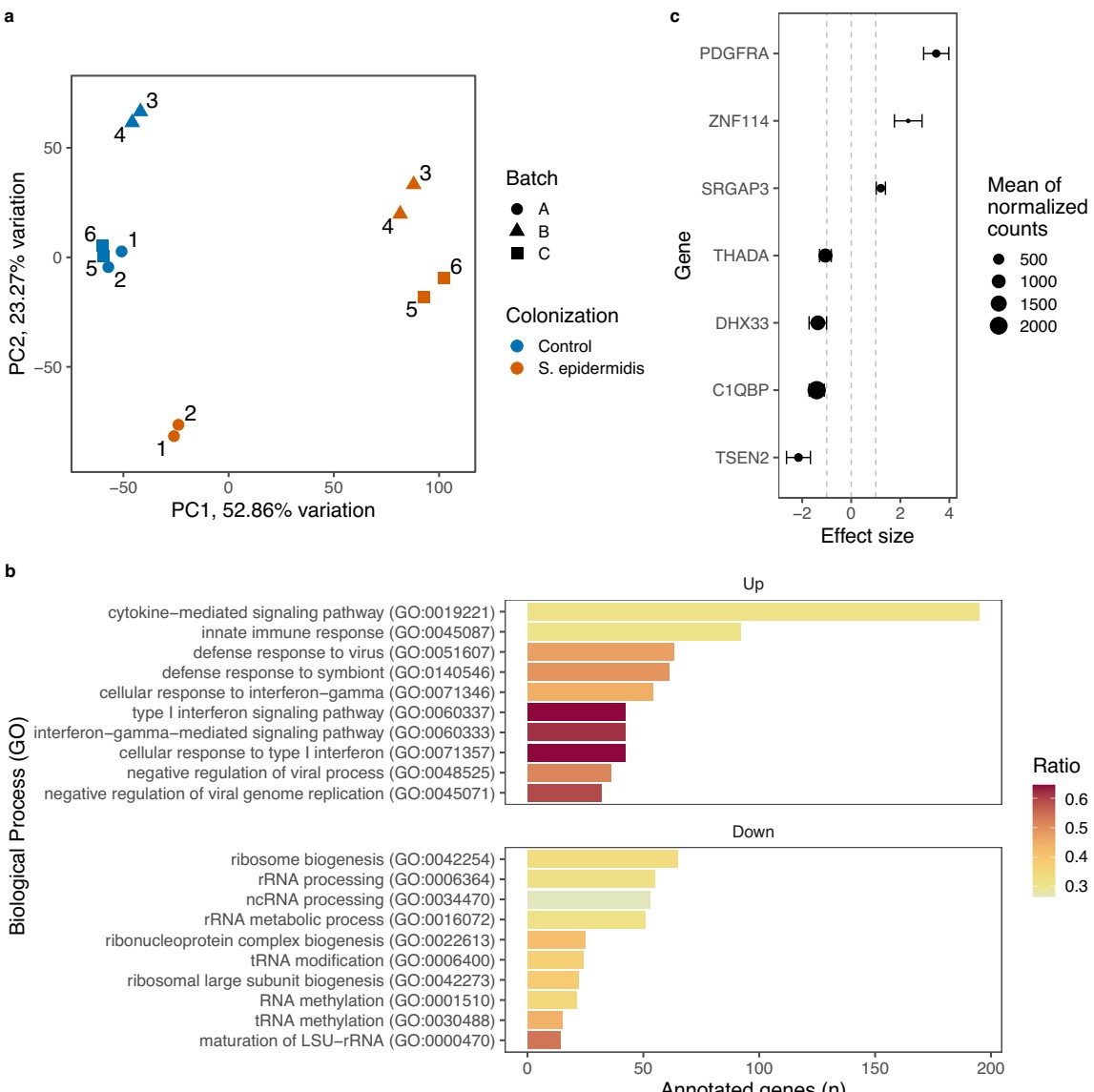

**Fig. 4 | GWAS genes expressed by keratinocytes co-cultured with *Staphylococcus epidermidis*. a** Bacteria were added to two-dimension keratinocyte cultures, which were cultivated in six replicates, two per weekly batch. First and second dimensions of principal component analysis of gene expression after variance stabilizing transformation (vst) are shown. Differential expression analysis was performed to compare the expression of keratinocyte genes in culture with and without *S. epidermidis*. **b** Enrichment of biological processes mapped to Gene Ontology (GO) was performed on differentially expressed genes [$q < 0.05$ (derived from Wald test on negative binomial generalized linear models) and absolute logarithmic (log2) fold change >1]. Top ten lowest adjusted *P* values (Fisher exact test) of each up and down regulated processes are shown, ordered by number of detected genes. Large subunit ribosomal ribonucleic acid (LSU-rRNA), transfer RNA (tRNA) and noncoding RNA (ncRNA) are abbreviated. **c** Change in the transcription of GWAS selected genes which were differentially expressed are shown. Approximate posterior estimation for generalized linear model (apeglm[78]) shrinkage was applied to effect size (log2 fold change). Error bars represent posterior standard deviation.

effect direction ($\beta_{Moist} = 8.8 \times 10^{-04}$; $\beta_{Dry} = 1.1 \times 10^{-03}$). Additional results suggested involvement of *Staphylococcus* ASVs in psoriasis (ASV012 [*Staphylococcus hominis*], $\beta = 4.0 \times 10^{-04}$), seborrhoeic keratosis (ASV010 [*Staphylococcus* (uncl.)], $\beta = 7.2 \times 10^{-04}$) and vitiligo (ASV012 [*Staphylococcus hominis*], $\beta = 1.2 \times 10^{-04}$). Potential protective effects of staphylococci in allergic rhinitis were also suggested (ASV012 [*Staphylococcus hominis*], $\beta = -1.1 \times 10^{-03}$). It is noteworthy that these are likely coagulase-negative staphylococci, which are typical members of the skin microbiota[43]. However, the ASV-level signals from the MR were only weakly or inconclusively supported by the sensitivity analysis (see Methods, Supplementary Data 5). Together, our findings suggest that members of the microbiota may modulate the health-disease balance in skin.

In summary, we conducted the first genome-wide association analysis dedicated to the human skin microbiota and identified 23 genome-wide significant loci. The combination of samples from different skin microenvironments of participants from two independent German cohorts allowed for robust results, despite the rather small number of included participants. The candidate genes have functions related to innate immune signaling, environmental sensing, cell differentiation, proliferation and fibroblast activity. Keratinocyte cultures challenged with a laboratory strain of *S. epidermidis* indicated regulation of seven candidate genes identified by GWAS, providing preliminary evidence that GWAS selected genes may be transcriptionally regulated by skin commensals. MR analysis further supported that specific skin microbiota features might have causal roles in the

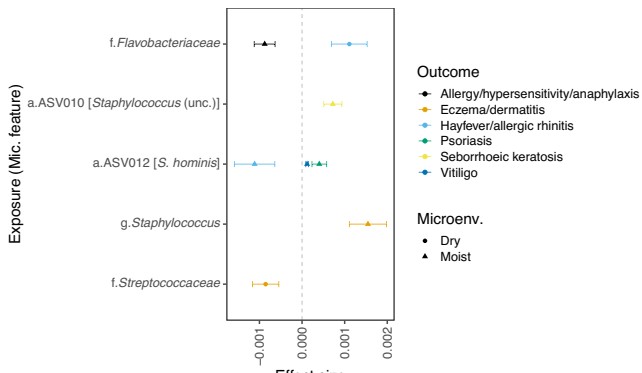

**Fig. 5 | Results from Mendelian Randomization analysis.** All exposure-to-outcome pairs with $q_{(trait)}$ <0.05 in the inverse-variance weighted 2-sample MR are shown. Error bars represent standard error. Microbial features are prefixed with their level, amplicon variant sequence (a.), genus (g.) and family (f.). Additional details and sensitive analyses can be found in Supplementary Data 5.

development of atopic dermatitis, but also suggested modulation of other non-infections skin diseases.

It needs to be considered that, despite our efforts to integrate information from different molecular levels and databases to understand the exact mechanisms by which the variants influence candidate gene function(s) and or expression and how this influences the skin microbiome, further and advanced experiments are needed. Likewise, it would be important to systematically establish differences in cutaneous gene expression with skin type, skin physiology and across age groups. Nevertheless, our results suggest a close interaction of the host genetic makeup and associated skin microbiomes. Furthermore, our findings point to the skin microbiota as a target for disease prevention and management, with potential for the development of personalized treatments for non-infectious, inflammatory skin conditions.

## Methods
### Cohorts' description, genotyping, imputation and harmonization
PopGen cohort participants were randomly recruited via the local population registry in Kiel, Germany, and as blood donors of the University Hospital Schleswig-Holstein, Campus Kiel[9]. Genotypes derived from the Affymetrix Genome-Wide Human single nucleotide polymorphism (SNP) Array 6.0 were quality controlled following a previously established protocol[44] and using the IKMB GWAS Quality Control Pipeline (https://github.com/ikmb/gwas-qc). Briefly, variants with excess missing data (>2%) and/or that deviated from Hardy-Weinberg Equilibrium [HWE, False Discovery Rate (FDR)[45] P value <10⁻⁵] were excluded. Samples with high missing data (>2%), high overall increased/decreased heterozygosity rates (i.e., ±5 standard deviation from the sample mean) and related individuals with a PLINK[46] PI_HAT score >0.1875 were removed. To assess population structure, we performed a principal components analysis (PCA) including individuals of the 1000 Genomes Phase3 ref.[47] and removed outlier individuals not matching a European ancestry. Imputation was performed with the Michigan Imputation Server[48] (Reference Panel: HRCr1.1 2016 (GRCh37/hg19); Array Build: GRCh37/hg19; rsq filter: off; Phasing Eagle 2.4 (phased output); Population: EUR; Mode: Quality Control & Imputation) and was followed by removal of monomorphic variants. These steps were performed following the miQTL cookbook instructions (https://github.com/alexa-kur/miQTL_cookbook#chapter-2-genotype-imputation).

KORA FF4 cohort participants from the youngest age group (39-48 years) that were previously genotyped as part of KORA S4 Survey were recruited from the southern German city of Augsburg and its two surrounding counties[8]. Genotyping and genotyping imputation were

performed by the KORA Study Center. Briefly, genotypes were derived from the Affymetrix Genome-Wide Human SNP Array 6.0 (KORA F4). Samples with missing data (>3%), mismatch with phenotypic and genetic gender and high heterozygosity rates (i.e., ±5 standard deviation from the sample mean) were removed. Samples were also checked for European ancestry, population outliers and compared with other existing genotype data of the same individual within the KORA cohort. Variants with excess missing data (>2%), deviating from HWE (P value <5×10⁻¹⁰) and Minor allele frequency (MAF) (<2%) were removed. Prephasing was done with SHAPEIT v2[49] and imputation with IMPUTE v2.3[50] (reference panel: 1000 Genomes Phase 3 integrated variant set release in NCBI build 37).

To harmonize both genotype datasets, resulting VCF (PopGen) and IMPUTE output (KORA FF4) files were converted to PLINK format using PLINK v1.9[46]. Participants that had their skin microbiota profiled (see section below) were selected and variants with MAF < 5% were removed. Genotype Harmonizer v 1.4.23 was used to update the KORA FF4 allele reference based on the PopGen data. SNPs with missingness >10% and non-biallelic SNPs were removed from PopGen data using PLINK v2.0-alpha-avx2-20200217. PopGen SNPs were references to set alleles in KORA F4 data, which also underwent removal of variants with missingness <10% and non-biallelic SNPs. Data sets were merged into PLINK files using PLINK v1.9 which contained SNPs available in both cohorts. Lastly, a principal component analysis (PCA) was produced with PLINK v1.9 to summarize the genetic population structure.

Written informed consent was obtained from all study participants. All protocols were approved by the ethics committees of the Medical Faculty of Kiel University (PopGen) and of the Bavarian Medical Association (KORA). We have complied with all relevant ethical regulations.

### Sampling collection and microbial profiling
Skin microbiota was sampled as described previously[3]. Briefly, skin swabs were taken with Catch-All Sample Collection Swab (Epicentre Biotechnologies, Madison, WIS) soaked in specimen collection fluid (SCF-1) from 4 cm² area of the skin site. Skin sites were selected to represent moist skin (antecubital fossa in both PopGen and KORA FF4), sebaceous skin (retroauricular fold in KORA FF4 and forehead in PopGen), and dry skin (volar and dorsal forearm in PopGen). Skin swabs were stored at -80 °C until DNA extraction using the QIAamp UCP Pathogen Mini Kit on an automated QIAcube system (QIAGEN GmbH, Hilden, Germany) for PopGen and the PowerSoil DNA Isolation Kit (MoBio Laboratories, Carlsbad, CA) for KORA FF4.

Bacterial profiles were based on the V1 and V2 variable regions of the gene coding for 16 S ribosomal RNA (rRNA). Briefly, V1-V2 regions were amplified with PCR performed with the primer pair 27F-338R. Pooled amplicon libraries were sequenced with MiSeq Reagent Kit v3 on the Illumina MiSeq (Illumina Inc., San Diego, CA). Sequencing reads were processed with DADA2 v1.10[51], resulting in an amplicon sequence variant (ASV) table, which records the number of times each exact ASV was observed in each sample[52]. ASV is a finer scale analogue of the operational taxonomic unit (OTU), which resolves the sequenced region variant down to a single-nucleotide difference level. ASVs were taxonomically classified down to genus level using RDP classifier algorithm based on Ribosomal Database Project (RDP) version 16 release with 50% confidence[53,54]. Species-level annotations were added to ASV sequences based on exact matches to the RDP database, using the function addSpecies() from DADA2 R package. Species-level abundances were not considered in the GWAS, as these are likely incomplete and possibly inaccurate[55], however annotations can still serve as proxies for sub-genus level placement of ASVs. Therefore, their species-level annotations were carried as part of the ASV annotation throughout the manuscript using square brackets, i.e. ASV001 [*Propionibacterium acnes*] or ASV001 [*P. acnes*]. Finally, sequences

were filtered to remove chloroplasts, mitochondria and low abundant ASVs (less than 0.1% of total sequence counts of a given skin site). Samples were removed if taken from a site with apparent skin abnormality or in which corticosteroids or antibiotics were applied in the last seven days before collection. Microbiota data was manipulated in R 3.6.2 using the Phyloseq package v1.34.0[56,57]. Details on sequencing, read processing and ASV filtering are provided in our previous study with the same dataset[3]. Finally, only samples from participants with genotype data were kept for downstream analysis.

### Association of microbial features with host SNPs

The association of SNPs was tested for multivariate (for inferences on the bacterial community; beta diversity), and univariate microbial features (for inferences on individual bacterial clades). Beta diversity was inferred from Bray-Curtis dissimilarities of rarefied amplicon variant (ASV) table (5,000 sequences per sample), calculated in R version 3.6.2 using the Vegan package v2.5-5[58]. Bacterial clades included ASVs and taxonomic groups ranging from genus to phylum. Taxonomic groups were obtained by merging the ASV sequence counts that had the same taxonomy at a certain rank, using the Phyloseq function tax_glom(). For each skin site, univariate features with a median sequence count higher than 50 and that were present in more than 100 participants were kept. In addition, univariate features were kept only if present in both sites of the same skin microenvironment, i.e., moist, sebaceous or dry. This effort resulted in 103 bacterial clades. To avoid redundancy, these clades were clustered together based on a 0.985 Spearman correlation cut-off. Clustering of clades were performed in each skin microenvironment separately because skin microenvironments have distinct bacterial profiles[3]. This effort resulted in a total of 79 bacterial features to be tested: 3 phyla, 4 classes, 7 order, 7 families, 15 genera and 43 ASVs.

Statistical tests were conducted for each microbial feature in each skin site from a single cohort following the framework established previously[7]. Because this process generates subsets of the whole data, additional variant inclusion criteria were implemented when necessary prior association tests. Accordingly, genetic variants were filtered (MAF > 5%) and coded into numeric features (0 = homozygous for reference allele; 1 = heterozygous; and 2 = homozygous for alternative allele). Only non-monomorphic variants were considered for testing. All tests were performed on the alternative allele as effect allele.

For tests with multivariate microbial features, distance-based redundancy analysis was performed with the vegan function capscale() with age, sex, BMI and the first ten genetic principal components (PCs) as covariates. The variables were selected because they were found as main confounders of the skin microbiota[3] and to account for the influence of the genetic background. The variance left unexplained by these covariates was extracted using the R residuals() function. The effect of genetic variants was estimated from the residual matrix with a distance-based F-test using moment matching[59]. For tests with univariate microbial features, zero-truncated non-rarefied count abundances were used. Outliers were filtered based on rarefied counts to account for uneven sequencing depths between samples. Samples were considered outliers when they deviated more than 5× the interquartile range (IQR) from the median abundance. Finally, count abundances (non-rarified) were fit with the Mvabund v4.1.6[60] function manyglm() in generalized linear models with negative binomial distributions and the covariates above mentioned as predictors. The logarithm of the total sequence counts of each sample was used as offset. Unexplained variance was extracted using residuals() function, which extracts from manyglm() models residuals that are normal[61]. The effect of genetic variants was estimated from the residuals using linear model. P value was calculated using the R summary function, which performs a two-sided t-test.

### Microenvironment-wise meta-analysis

Genomic inflation ($\lambda_{GC}$) was calculated for all tests using the regression method as implemented in GenABEL v1.8-0 R package[62]. All values were below 1.02, indicating no genomic inflation. Because skin microbiota profiles are distinctive between microenvironments[3], meta-analyses were performed combining data sets that originated from skin sites of the same microenvironment. Therefore, results from moist skin sites were merged into one meta-analysis and results from sebaceous skin sites into another, because skin sites from these microenvironments are from different cohorts. Because the distance-based F-test applied to multivariate features do not produce beta values, a fixed effect meta-analysis was performed with METAL release 2011-03-25[63], with meta-analysis P values ($P_{Meta}$) and sample size based weighting. For univariate features, an inverse-variance weighted fixed effect meta-analysis was performed with METASOFT v2[64] on beta values and their standard errors. Meta-analysis results were reported significant if genome-wide significance ($P_{Meta} < 5 \times 10^{-8}$) was achieved and the association was found nominally significant in the two skin sites ($P < 0.05$). Because samples from the two dry skin sites came from a single cohort (PopGen), results were combined and considered significant if the P value of at least one skin site was genome-wide significant ($P < 5 \times 10^{-8}$) with at least nominal significance at the other skin site ($P < 0.05$). In this case, the lowest P value was reported as the $P_{Meta}$ value. The study-wide significance threshold was calculated considering the number of microbiota features tested ($P_{Meta} < 5 \times 10^{-8}/80 = 6.3 \times 10^{-10}$).

### Fine-mapping and gene prioritization

Genes were considered of interest when containing potentially causal variants and/or these variants were significantly associated with the gene expression in skin tissue. Fine-mapping was performed to explore the most likely causal set of variants using shotgun stochastic search algorithm implemented in FINEMAP v1.4[65]. For moist and sebaceous microenvironments, fine-mapping was performed with summary statistics (beta values and their standard errors) from meta-analysis. For dry microenvironment, beta values and their standard errors from volar forearm were used for fine-mapping. Genes were reported when intersecting with the range of the 95% posterior credible SNP set assuming one causal variant as input parameter for the algorithm. If fine-mapping did not find a credible set (<50 variants), or for beta-diversity results, genes with variants with LD > 0.6 to the lead SNP were reported. SNPs and genes were annotated using the R package biomaRt v2.48.0[66].

To investigate whether genetic variants could affect gene expression in skin tissues, prioritized variants selected by fine-mapping or in LD > 0.6 were mapped to Genotype-Tissue Expression (GTEx) Project[10] database v8 (lower leg and suprapubic skin tissues). Briefly, chromosomal positions in genome assembly hg19 (GRCh37) were converted to hg38 (GRCh38) using LiftOver from the human genome browser at the University of California Santa Cruz (UCSC)[67]. These positions were then mapped to single-tissue cis-quantitative trait locus (QTL) data downloaded (11/06/2021) from the GTEx portal[10], specifically the file GTEx_Analysis_v8_eQTL.tar, which contains genes of which expression are significantly associated with genetic variants based on permutations. Only data from skin tissues (suprapubic non-sun-exposed and lower leg sun-exposed) were used in this analysis.

### Expression of genes in skin tissues and cell types

Consensus transcriptional expression of genes in skin tissue were retrieved from the Human Protein Atlas version 20.1[11], which additionally includes data sets from the GTEx[10] and the Functional annotation of the mammalian genome (FANTOM5)[68] projects. Single-cell RNA-Seq data of skin from healthy individuals ($n = 5$) were retrieved from a recent study by Solé-Boldo et al.[12].

## Mendelian randomization

Mendelian randomization (MR) was performed using summary statistics of univariate association analyses as 'exposures' and six selected skin-related traits as 'outcomes' (allergy/hypersensitivity/anaphylaxis, seborrheic keratosis, eczema/dermatitis, hay fever/allergic rhinitis, psoriasis, vitiligo). Outcome summary statistics were retrieved from UK Biobank using the R package TwoSampleMR v0.5.5[69]. UK Biobank originated from the IEU Open GWAS Project database[70]. All variants in the microbial 'exposures' with an association $P$ value $<10^{-5}$ were included in the analyses. After harmonization with the exposure data, only independent variants were retained using the clump_data() function with default parameters. Additionally, variants with an F statistic $<10$ were excluded from the analysis to avoid weak instrument bias[71]. In case of more than two independent retained variants, inverse variance weighted (IVW) MR analysis was performed as primary analysis, otherwise Wald-ratio was calculated. For exposures with more than two instrument variables, weighted mean, and MR Egger regression were performed for sensitivity analysis. MR Egger regression with non-significant beta values ($P > 0.05$) and weighted median MR results with significant ($P < 0.1$) and concordant effect direction to the IVW MR analysis were regarded as supporting. $P$ values of the primary MR analysis (IVW or Wald-ratio) were corrected for multiple testing using per-trait and global FDR correction[36]. All MR analyses were conducted in R v3.6.1.

## Keratinocytes co-culture with *Staphylococcus epidermidis*

Normal human epidermal keratinocytes (NHEKs) (foreskin of a 0-year-old male Caucasian donor; Promocell, Heidelberg, Germany, Lot number 407Z001) were cultured in Keratinocyte Growth Medium (KGM; Lonza Biosciences, Walkersville, USA) + supplements + $CaCl_2$ + penicillin/streptomycin at 37 °C and 5% $CO_2$. Cells were used at passages 4-6. Keratinocytes were seeded into 6-well plates and grown until confluency. *Staphylococcus epidermidis* (Winslow and Winslow) Evans (ATCC 14990) was cultured in Tryptic Soy Broth (TSB; Thermo Fischer, Waltham, USA) medium at 37 °C. For co-cultivation with keratinocytes, bacteria were centrifuged, resuspended in KGM + $CaCl_2$, and added to the keratinocytes at an optic density (OD) of 0.1 in KGM + $CaCl_2$. A total of six replicates of each condition, with and without the addition of *S. epidermidis*, were performed, two replicates per weekly batch. Plates were centrifuged at 350 x g for 5 min to allow bacteria to settle on the bottom. After 3 h incubation, plates were washed, and KGM + $CaCl_2$ with gentamycin was added for a further incubation of 23 h at 37 °C and 5% $CO_2$. Plates were washed twice before RNA isolation using Trizol (Thermo Fischer, Waltham, USA) as per manufacturer's instruction.

Sequencing libraries were prepared using TrueSeq Stranded mRNA kit (Illumina Inc., San Diego, USA). Sequencing was performed on the Illumina NovaSeq 6000 platform (Illumina Inc., San Diego, USA) with 2 × 50 base pairs length. Raw sequences were processed using the nf-core/rnaseq pipeline v3.0[72,73], which includes adapter quality trimming with Trim Galore (https://github.com/FelixKrueger/TrimGalore), removal of ribosomal RNA with SortMeRNA[74], alignment with STAR[75] and transcript quantification with Salmon[76]. The human genome assembly hg19 was used as reference. Differentially expressed genes were detected with the R package DESeq2 v.1.30.0[77]. Wald test was performed with negative binomial generalized linear models, which included the weekly batch and whether *S. epidermidis* was added to the culture or not (~batch + condition). $P$ were corrected for multiple testing using the FDR method[45]. Approximate posterior estimation for generalized linear model (apeglm[78]) shrinkage was applied to logarithmic (log2) fold change (LFC). Results were considered significant based on $q$ values ($<0.05$) and LFC (absolute LFC $> 1$). Enrichment of expressed genes up and down regulated were performed using the R package enrichR v3.0 and the GO_Biological_Process_2021 database[79]. Enriched pathways were considered significant based on $q$ values ($<0.05$; Fisher exact test). To get an overview of the effect of the *S. epidermidis* addition to keratinocyte cultures, transcriptional profiles were visualized through principal component analysis (PCA). First, variance stabilizing transformation (VST) from the R package DESeq2 v.1.30.0[77] was applied to the transcriptional data. PCA was performed as implemented in the R package PCAtools v. 2.4.0[80].

## Reporting summary

Further information on research design is available in the Nature Research Reporting Summary linked to this article.

## Data availability

Raw 16 S rRNA gene amplicon sequences of PopGen participants were deposited at the European nucleotide archive (ENA) under accession code PRJEB41215. GWAS summary statistics generated in this study are available at GWAS catalogue under accession codes GCST90133164-GCST90133313. Phenotype data from PopGen individuals can be accessed through the Material Data Access Form from the PopGen Biobank (Schleswig-Holstein, Germany). Information about the Material Data Access Form and how to apply can be found at http://www.uksh.de/p2n/Information+for+Researchers.html. KORA data are available at https://www.helmholtz-munich.de/en/kora/for-scientists/cooperation-with-kora/index.html upon request by means of a project agreement. In addition, the following public database and resources were used: 1000 Genomes Phase3 ref. 47, Ribosomal Database Project (RDP) version 16[54], Genotype-Tissue Expression (GTEx) Project database v8[10], Functional annotation of the mammalian genome (FANTOM5)[68], Skin single-cell data from by Solé-Boldo et al.[12], UK Biobank and the IEU Open GWAS Project database[70].

## Code availability

Code used in the analysis are available at https://github.com/LucasMS/skin.mgwas.pub[81].

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

## Acknowledgements

We are grateful to all participants and study staff from the Biobank PopGen (Dr Gunar Jacobs and team) and the KORA Studienzentrum (Dr Margit Heier and team). We thank the staff from UKSH Dermatology laboratory (particularly, Anke Rose), the IKMB microbiome laboratory, the IKMB DNA laboratory and the IKMB sequencing laboratory for their excellent support. We are grateful to Dr Sören Franzenburg and Eike Matthias Wacker for assistance and troubleshooting. We thank Martin Schulzky for the design of skin site icons. The project leading to this application has received funding from the Deutsche Forschungsgemeinschaft (DFG) Grant no. WE2678/14-1 (granted to S.W.) and the Innovative Medicines Initiative 2 Joint Undertaking under Grant Agreement no. 821511 (BIOMAP, granted to S.W.). This Joint Undertaking receives support from the European Union's Horizon 2020 research and innovation programme and EFPIA. This publication reflects only the author's view and the JU is not responsible for any use that may be made of the information it contains. This work was also supported by the Deutsche Forschungsgemeinschaft (DFG) Collaborative Research Center 1182 'Origin and Function of Metaorganisms' (grant no. SFB1182, Project A2 granted to A.F.). The study received infrastructure support from the DFG research unit "miTarget" (Projektnummer 426660215; EL 831/5-1 granted to A.F.). The KORA study was initiated and financed by the Helmholtz Zentrum München – German Research Center for Environmental Health, which is funded by the German Federal Ministry of Education and Research (BMBF) and by the State of Bavaria.

## Author contributions

Study was designed by L.M.S., F.K., E.R., H.E., F.U.W., D.E., A.F., S.W. and M.C.R. Sample, data collection and processing were done by E.R., H.E., L.T., W.L., C.G., A.P., C.B. Analysis was performed by L.M.S., F.D., H.E., S.J., L.M., F.U.W., D.E., N.S., H.B. and M.C.R. Manuscript was written by L.M.S., F.D., E.R., H.E., F.U.W., A.F., S.W. and M.C.R. All authors revised the manuscript.

## Funding

## Competing interests

The authors declare no competing interests.

## Additional information

Lucas Moitinho-Silva [1,2], Frauke Degenhardt[1], Elke Rodriguez [2], Hila Emmert [2], Simonas Juzenas [1,3], Lena Möbus[2],
Florian Uellendahl-Werth [1], Nicole Sander[2], Hansjörg Baurecht[4], Lukas Tittmann[5], Wolfgang Lieb [6],
Christian Gieger [7,8], Annette Peters [7], David Ellinghaus [1], Corinna Bang[1], Andre Franke [1,9] ✉,
Stephan Weidinger [2,9] ✉ & Malte Christoph Rühlemann [1,9]

[1]Institute of Clinical Molecular Biology, Kiel University, Kiel, Germany. [2]Department of Dermatology and Allergy, University Hospital Schleswig-Holstein,
Kiel, Germany. [3]Institute of Biotechnology, Life Science Centre, Vilnius University, Vilnius, Lithuania. [4]Department for Epidemiology and Preventive Medicine,
University of Regensburg, Regensburg, Germany. [5]Biobank PopGen and Institute of Epidemiology, Kiel University, Kiel, Germany. [6]Institute of Epidemiology,
Kiel University, Kiel, Germany. [7]Institute of Epidemiology, Helmholtz Zentrum München – German Research Center for Environmental Health,
Neuherberg, Germany. [8]Research Unit of Molecular Epidemiology, Helmholtz Zentrum München – German Research Center for Environmental Health,
Neuherberg, Germany. [9]These authors contributed equally: Andre Franke, Stephan Weidinger, Malte Christoph Rühlemann. ✉e-mail: a.franke@mucosa.de;
sweidinger@dermatology.uni-kiel.de

