## [Peer Review File · Nature Communications]

Host genetic factors related to innate immunity, environmental sensing and cellular functions influence human skin microbiotaREVIEWER COMMENTS

Reviewer #1 (Remarks to the Author):

Moitinho-Silva investigated the effect of genetics on skin microbiota across three different skin microenvironments through meta-analyses of genome-wide association studies (GWAS) of two population-based German cohorts. Thirty genome significant loci were identified. The authors report that genetic variants associated with the skin bacteria were localized in genes related to pathogen sensing and regulation of response to pathogens. Mendelian randomization revealed association with a number of inflammatory diseases and certain microbial pathogens.

Comments:

The authors have been appropriately stringent in reporting genome-wide significant loci.

The description of SNPs and nearby genes needs cleaning up. For example it is simplistic to assume that all SNPs e.g. in intronic regions, are affecting regulation of the closest gene – the classic example of an intronic SNP affecting the next gene is in the case of lactase persistence. Hence, the authors should temper their description of associated genes, and also look nearby to see if there are better candidates. SNPs may also have an effect at more than one gene per associated locus. Moreover, in some instances it isn't clear why they have focused on a single gene (e.g. HTT) when the lead SNP lies in a gene desert between two different genes. In other instances, such as the case of the SNP within C1QBP, the authors discuss DHX33 in the next section, but it isn't clear from what is written that DHX33 lies within the same locus as C1QBP. The selection of genes from the GWAS also impacts on results presented in Table 1 and supplementary Table 3.

Co-culturing of keratinocytes with staphylococcus led to changes in expression of 15 transcripts of genes reported by the authors in the associated loci. It is not clear what to make the results of this experiment. It would be helpful to know how many co-cultures were performed, how they differed following a principal component analysis and to be provided with all transcripts with $q < 0.05$ as a supplementary table (and not just those identified from their GWAS). The authors should also perform pathway analyses which could lend more credence to their approach and findings pertaining to their top genes.

Minor comments:

Typo p9: relative

Grammar p23: correct grammar "genes which expression is"

Supplementary p30: Here the authors are presenting data on transcripts not genes. Please correct where the incorrect term has been used throughout the manuscript as well.

Were fetal or adult keratinocytes used for the co-cultures?

Reviewer #2 (Remarks to the Author):

In this very interesting study, Moitinho-Silva et al conducted meta-analysis of GWAS from two German cohorts and correlated this with 16S rRNA sequencing data to address potential mechanisms by which genetic variation influences the shape of the skin microbiome. The authors further investigated if keratinocyte expression of genes identified by the GWAS were influenced by *S. epidermidis*. The topic of the manuscript is potentially important for gaining a better understanding of the host genetic factors that shape the microbial community on the skin. However, the approach makes it difficult to assess how the identified SNPs influence gene function or expression and there is no data to support a direct mechanism by which these SNPs can promote the observed dysbiosis.

Major Comments:

1) The authors used the term "microbial future" in the manuscript but the definition of this term is vague and does not clearly describe the changes occurring in the microbial community.

2) The topical skin microbial community is different depending on body site and age even within individuals. Please explain how the performed cross-sectional study comparing "microbial future" to an individual's SNP variants is biologically significant to allow for speculation on the influence of genetic variation of a specific gene on small representative populations of the microbial community.

3) 16S sequencing data only provides the proportion of each taxa, not absolute abundance. Speculation by the authors that the observed gene variations influence the abundance of members of microbial community is therefore not appropriate.

3) The authors identified a higher genome-wide association to microbial future in Moist > Dry > Sebaceous areas. Are the genes identified by GWAS analyses differentially expressed in these 3 skin sites?

4) Taxonomy was assigned to a ~300bp fragment of the V1-V2 region of the 16S amplicon at the order level to the ASV level. There is a serious concern about the accuracy of the taxonomical assignment based on this approach. Further analysis with deeper sequencing to determine species level assignments is necessary.

5) Stimulation with *S. epidermidis* changed expression of 15 genes in human keratinocytes out of the 31 genes identified by the GWAS approach. However, the gene expression data did not support the proposed mechanism of how the localization of SNPs in each gene influences the microbial community.

6) Current research has shown that the immunomodulatory effect of *S. epidermidis* is highly dependent on strain. In addition, there is a large strain level variation within the community of *S. epidermidis* within the skin microbiome. In contrast, the authors utilized only a single strain of *S. epidermidis* for stimulation of keratinocytes to validate the GWAS analyses.

7) The authors should discuss how the localization of each SNP potentially influences the function/ expression of the genes discussed.

Minor

8) Lines 235, 251, 254 and 259: Parentheses are empty.

Dear reviewers,

We thank your dedicated review of our manuscript. We appreciate your insightful comments and have attempted to respond to each one of them. Based on your comments, we made important changes in the manuscript, which we think has much improved. Below you will find our reply to your points.

Best regards,

Lucas Moitinho-Silva

REVIEWER COMMENTS

Reviewer #1 (Remarks to the Author):

Moitinho-Silva investigated the effect of genetics on skin microbiota across three different skin microenvironments through meta-analyses of genome-wide association studies (GWAS) of two population-based German cohorts. Thirty genome significant loci were identified. The authors report that genetic variants associated with the skin bacteria were localized in genes related to pathogen sensing and regulation of response to pathogens. Mendelian randomization revealed association with a number of inflammatory diseases and certain microbial pathogens.

Comments:

The authors have been appropriately stringent in reporting genome-wide significant loci.

We appreciate the reviewer's comments and the acknowledgment of the appropriate framework of reporting our results.

The description of SNPs and nearby genes needs cleaning up. For example it is simplistic to assume that all SNPs e.g. in intronic regions, are affecting regulation of the closest gene – the classic example of an intronic SNP affecting the next gene is in the case of lactase persistence. Hence, the authors should temper their description of associated genes, and also look nearby to see if there are better candidates. SNPs may also have an effect at more than one gene per associated locus. Moreover, in some instances it isn't clear why they have focused on a single gene (e.g. HTT) when the lead SNP lies in a gene desert between two different genes. In other instances, such as the case of the SNP within C1QBP, the authors

discuss *DHX33* in the next section, but it isn't clear from what is written that *DHX33* lies within the same locus as *C1QBP*. The selection of genes from the GWAS also impacts on results presented in Table 1 and supplementary Table 3.

We agree with the reviewer that prioritizing genes solely based on location would not be appropriate. Therefore, we used the following two criteria to select genes of interest:

- Overlap with a potentially causal SNP. A SNP was considered potentially causal when identified by fine-mapping or linkage disequilibrium with the lead SNP >0.6 .
- Gene expression is significantly associated with a potentially causal SNP in skin tissue based on GTEx portal¹⁰ data.

We apologize that our approach was not clearly outlined in the previous version of the manuscript. We now explain it more clearly both in the results (“A total of 23 genes were found of interest for containing potentially causal variants and/or because these variants were significantly associated with the gene expression in skin tissue from the GTEx portal¹ (Table 1)”) and methods section (“Genes were considered of interest when containing potentially causal variants and/or these variants were significantly associated with the gene expression in skin tissue.”).

Furthermore, we now make it clear why we focused on some genes. In the results section we now state that we further explored genes of interest that have a potential functional role related to the host-microbiome interface”, and added the conclusion: “Keratinocyte proliferation, differentiation and function as well as innate immune signaling are major forces contributing to the complex function of the skin barrier. Therefore, it is conceivable that the discovered GWAS associations may represent links between the skin barrier and members of the skin microbiota.”

Finally, we also outlined that *DHX33* and *C1qBP* genes belong to the same locus.

Co-culturing of keratinocytes with staphylococcus led to changes in expression of 15 transcripts of genes reported by the authors in the associated loci. It is not clear what to make the results of this experiment. It would be helpful to know how many co-cultures were performed, how they differed following a principal component analysis and to be provided with all transcripts with $q < 0.05$ as a supplementary table (and not just those identified from their GWAS). The authors should also perform pathway analyses which could lend more credence to their approach and findings pertaining to their top genes.

We thank the reviewer for these insightful suggestions, which we followed when revising and expanding the corresponding section, which now includes the description of the replicates, a principal component analysis showing the overall transcriptional response patterns and a pathway enrichment analysis based on GO biological processes. We agree with the reviewer that this allows a more comprehensive interpretation of our co-culturing experiments, and that now the expression of the GWAS selected genes is put into a clearer context.

We decided to also include a fold change filter to enrich our result with genes that most likely have biological significance. In addition to statistical significance ($q < 0.05$), all genes listed display an at least 2-fold up- or down-regulation to be considered differentially expressed (**absolute log₂ fold change > 1**).

This more stringent approach reduced the number of GWAS selected differentially expressed genes from 15 to 7.

Minor comments:

Typo p9: relative

Corrected

Grammar p23: correct grammar "genes which expression is"

Corrected

Supplementary p30: Here the authors are presenting data on transcripts not genes. Please correct where the incorrect term has been used throughout the manuscript as well.

We rectified the incorrect usage and apologize for any possible confusions.

Were fetal or adult keratinocytes used for the co-cultures?

Keratinocytes were derived from the foreskin of a 0-year-old male donor. This is now stated in both results and methods sections.

Reviewer #2 (Remarks to the Author):

In this very interesting study, Moitinho-Silva et al conducted meta-analysis of GWAS from two German cohorts and correlated this with 16S rRNA sequencing data to address potential mechanisms by which genetic variation influences the shape of the skin microbiome. The authors further investigated if keratinocyte expression of genes identified by the GWAS were influenced by *S. epidermidis*. The topic of the manuscript is potentially important for gaining a better understanding of the host genetic factors that shape the microbial community on the skin.

We appreciate the reviewer's positive feedback and constructive criticism.

However, the approach makes it difficult to assess how the identified SNPs influence gene function or expression and there is no data to support a direct mechanism by which these SNPs can promote the observed dysbiosis.

We agree with the reviewer that more data is desired to understand how the selected variants may influence gene function and the skin microbiome. To support the SNP-microbiome relations found in our GWAS approach, we integrated information from different molecular levels and databases, e.g.:

- Our strict selection of potentially causal variants and gene of interest maximized the likelihood of finding *bona fide* gene-microbiome associations (see answer to reviewer 1, comment 1 for details).
- We used single-cell and protein expression information from public data to assess the relevance of the candidate genes to the skin tissue.

- We performed *in vitro* co-culturing assays of *S. epidermidis* and keratinocytes to gain insights into the potential participation of the candidate genes to the host-cell-microbiome interaction.

However, we agree with the reviewer that additional advanced functional experiments will be needed to achieve a more detailed understanding of the interaction between the variants of interest, gene function and/or expression, and microbiome composition. We now acknowledge both our efforts as well as the reviewer comment in the sentence below, added to the conclusion section:

“However, it needs to be considered that, despite our efforts to integrate information from different molecular levels and databases to understand the exact mechanisms by which the variants influence candidate gene function(s) and or expression and how this influences the skin microbiome, further and advanced experiments are needed. Likewise, it would be important to systematically establish differences in cutaneous gene expression with skin type, skin physiology and across age groups.”

Major Comments:

1) The authors used the term “microbial future” in the manuscript but the definition of this term is vague and does not clearly describe the changes occurring in the microbial community.

We apologize for being unclear. We now define the term “microbial feature” in the results section. It refers to relative abundances of ASVs and non-redundant taxonomic groups ranging from genus to phylum as well as multivariate community composition. Details on how multivariate community composition and univariate clade-abundances were calculated are found in the methods section. This approach describes the changes occurring in the microbial community, both at individual groups of members as well as the whole community.

2) The topical skin microbial community is different depending on body site and age even within individuals. Please explain how the performed cross-sectional study comparing “microbial future” to an individual’s SNP variants is biologically significant to allow for speculation on the influence of genetic variation of a specific gene on small representative populations of the microbial community.

Thank you for this question. We agree that many factors can have strong influence on the microbial community of the skin. These include host characteristics and environmental factors² as well as genetic factors. The latter have been previously suggested by a twin study where heritability accounted for more than 50% for some skin microbial traits³. The likelihood of biological significance of our observed associations between genetic variants and the skin microbiome was maximized by our rigorous statistical procedures and careful study design. In addition to the approaches described in the previous comment on how different layers of molecular information were integrated to support and give context to our findings, we:

- Accounted for the effect of age, gender and body mass index on skin microbiome variation.
- Accounted for variation between body sites by performing microenvironment-wise meta-analysis.
- Focused on population-based effects and not individual variations by performing meta-analysis on results on two independent cohorts.

However, we agree with the reviewer that the results from this first mGWAS on the skin certainly need to be corroborated and extended by additional experiments, as now stated in the conclusion section.

3) 16S sequencing data only provides the proportion of each taxa, not absolute abundance. Speculation by the authors that the observed gene variations influence the abundance of members of microbial community is therefore not appropriate.

We agree with the reviewer that sequencing-based assessments of microbial communities can – without additional efforts to quantify total microbial load – only give insights into the composition of the community, thus the relative abundances of the individual members, and not of the absolute abundances. We ensured that this impression is no longer given in the manuscript and refer to “relative abundances” or “count abundances” throughout the manuscript where this is appropriate.

3) The authors identified a higher genome-wide association to microbial future in Moist > Dry > Sebaceous areas. Are the genes identified by GWAS analyses differentially expressed in these 3 skin sites?

This is an interesting question. The answer of it would represent a step towards a better understanding the associations patterns observed in our mGWAS approach. Unfortunately, we do not have the experimental data that would allow for answering and are unaware of the existence of public data that could be used for this purpose. Because the answer to this question does not affect our main results and their current interpretation, we feel that the answer to this question is better suited as part of a future, follow-up study.

4) Taxonomy was assigned to a ~300bp fragment of the V1-V2 region of the 16S amplicon at the order level to the ASV level. There is a serious concern about the accuracy of the taxonomical assignment based on this approach. Further analysis with deeper sequencing to determine species level assignments is necessary.

We understand the reviewers concerns about annotations at low taxonomic levels. We now clarified our annotation approach and apologize for the lack of methodological detail in the previous version of the manuscript. We used a two-step method to derive taxonomic annotations.

1. To classify 16S rRNA gene amplicons sequences down to the genus level, we chose a widely used approach employing a naive Bayesian classifier. This method is rather robust to false annotation. However, it might lack sensitivity for some clades⁴. In this approach, the annotation quality is dependent on the database quality, which poses no source of concern in our study due to the combination of a relatively well characterized biome and the use of the curated RDP database.
2. Species-level annotations were performed by searching for **exact** matches of the ASV sequence to the database. While this approach – again of course depending on the database – can and will lead to highly specific species-level annotations for some sequences, the method is also sensitive to **exact** sequence variants simply not being present in the DB. For ASV annotation, this method, in our opinion, is appropriate⁵.

However, to some extent we share the reviewer’s concern. In fact, limited sensitivity of ASV annotation at the species level can lead to inaccurate relative abundances of species-level features presented in the

manuscript. For this reason, **we decided to remove all analyses carried out with species level microbial features from the manuscript.** Consequently, the overall number of tested features and the genome-wide significant signals were reduced, and loci 3, 8, 15, 16, 19, 21 and 30 were removed. Except for locus 15, all of these were in genomic areas without known coding sequences, thus the overall outcomes were not impacted.

We also want to emphasize that we fully agree with the reviewer that broad accurate species-level assignments for all taxonomic groups are desirable. Depending on the availability of sufficient high-quality biomaterial and funding in future studies we will aim for full-length 16S amplicons² or by shotgun metagenomics. However, as for now, we believe that the presented first microbiome GWAS for skin communities ever although based on the V1-V2 amplicon data is very informative and will serve as a starting point for future research.

5) Stimulation with *S. epidermidis* changed expression of 15 genes in human keratinocytes out of the 31 genes identified by the GWAS approach. However, the gene expression data did not support the proposed mechanism of how the localization of SNPs in each gene influences the microbial community.

In response to this and the reviewer 1 second comment, we have substantially expanded the corresponding section, which now puts the expression of GWAS candidate genes into a clearer biological context and added a sentence to the concluding section outlining limitations of our analysis, including the fact that our experimental data cannot precisely tell “how the localization of SNPs in each gene influences the microbial community.” (see above)

6) Current research has shown that the immunomodulatory effect of *S. epidermidis* is highly dependent on strain. In addition, there is a large strain level variation within the community of *S. epidermidis* within the skin microbiome. In contrast, the authors utilized only a single strain of *S. epidermidis* for stimulation of keratinocytes to validate the GWAS analyses.

We agree with the reviewer that the *S. epidermidis* symbiotic role is highly dependent on the strain characteristics. For this reason, we decided to perform our experiments with a well characterized *S. epidermidis* laboratory strain. Our experimental choice has the following advantages:

- The *S. epidermidis* ATCC 14990 strain is a well-known and characterized strain and can be easily obtained by other investigators, facilitating the interpretation and reproduction of our experiments.
- Despite being a laboratory strain, *S. epidermidis* ATCC 14990 is close to the strains found in our study participants as indicated by 100% overlap and match to the most abundant *Staphylococcus* ASV in our investigation.
- The *S. epidermidis* ATCC 14990 strain allows for a more general interpretation of the results in contrast to other, more functionally specific strains.

In vitro models that capture the broad spectrum of host-bacteria symbiosis are desired for better understanding the dynamics of this relation. This could be obtained by having a larger panel of skin microbial commensal strains as well as 3D skin cultures. However, we hope that the reviewer agrees that the battery of experiments needed for such an approach is somewhat out of the scope of the

current manuscript. We now acknowledge these points in the expanded version of the manuscript section “Expression of candidate genes by keratinocytes co-cultured with *Staphylococcus epidermidis*”, in which we openly discuss the strengths and potential shortcomings of our approach.

7) The authors should discuss how the localization of each SNP potentially influences the function/ expression of the genes discussed.

Thank you – may we please refer to our reply to comment #1.

Minor

8) Lines 235, 251, 254 and 259: Parentheses are empty.

Corrected

References

- 1 GTEx Consortium. The Genotype-Tissue Expression (GTEx) project. *Nat Genet* **45**, 580-585, doi:10.1038/ng.2653 (2013).
- 2 Moitinho-Silva, L. *et al.* Host traits, lifestyle and environment are associated with the human skin bacteria. *Br J Dermatol*, doi:10.1111/bjd.20072 (2021).
- 3 Si, J., Lee, S., Park, J. M., Sung, J. & Ko, G. Genetic associations and shared environmental effects on the skin microbiome of Korean twins. *BMC Genomics* **16**, 992, doi:10.1186/s12864-015-2131-y (2015).
- 4 Johnson, J. S. *et al.* Evaluation of 16S rRNA gene sequencing for species and strain-level microbiome analysis. *Nat Commun* **10**, 5029, doi:10.1038/s41467-019-13036-1 (2019).
- 5 Edgar, R. C. Updating the 97% identity threshold for 16S ribosomal RNA OTUs. *Bioinformatics* **34**, 2371-2375, doi:10.1093/bioinformatics/bty113 (2018).

REVIEWER COMMENTS

Reviewer #1 (Remarks to the Author):

The authors have responded appropriately to my concerns. I have no others.

Reviewer #2 (Remarks to the Author):

This manuscript has been slightly revised from the original by editing the paper to recognize the limitations of the approach and softening the conclusions. However, a key element of the main conclusions rest in the functional correlations with *S. epidermidis*. This important part of the paper remains incomplete, with only a superficial assessment based on ATCC14990, which, unlike the claims of the authors do not permit a general interpretation.

NCOMMS-21-44792A

Response to reviewers' comments

Reviewer #1 (Remarks to the Author):

The authors have responded appropriately to my concerns. I have no others.

We thank the reviewer for revisiting our manuscripts and agreeing with our answers to the comments made.

Reviewer #2 (Remarks to the Author):

This manuscript has been slightly revised from the original by editing the paper to recognize the limitations of the approach and softening the conclusions. However, a key element of the main conclusions rest in the functional correlations with *S. epidermidis*. This important part of the paper remains incomplete, with only a superficial assessment based on ATCC14990, which, unlike the claims of the authors do not permit a general interpretation.

We thank the reviewer for the suggestion to perform an in-depth functional analysis, and we absolutely agree that a series of sophisticated experiments using multiple strains of skin commensals and different models would certainly help to achieve a more detailed understanding of the interaction between the identified host gene variants, their function and/or expression, and microbiome composition. However, the focus of the current study was to examine genetic contributions to skin microbiome composition and we think that our study did yield interesting results in this context. Further, our admittedly preliminary functional analysis provides initial evidence that GWAS selected genes are indeed transcriptionally regulated by skin commensals. A more comprehensive characterization of the mechanisms by which the identified genome-wide significant host gene variants influence candidate gene function(s) and or expression and how this influences the skin microbiome is on our agenda but requires careful planning and considerable efforts. We think that this is somewhat outside the scope of this paper. We now more specifically outlined the limitations of our initial functional observations and the need for additional experiments, and hope that the reviewer agrees that the existing data is of interest to the scientific community and deserve publication also to stimulate further research.